# Illumination of a progressive allosteric mechanism mediating the glycine receptor activation

Sophie Shi[1,2,6], Solène N. Lefebvre [1,2,6], Laurie Peverini [1], Adrien H. Cerdan [3], Paula Milán Rodríguez [1,3], Marc Gielen [1,4], Jean-Pierre Changeux[5], Marco Cecchini [3] ✉ & Pierre-Jean Corringer[1] ✉

Pentameric ligand-gated ion channel mediate signal transduction at chemical synapses by transiting between resting and open states upon neurotransmitter binding. Here, we investigate the gating mechanism of the glycine receptor fluorescently labeled at the extracellular-transmembrane interface by voltage-clamp fluorometry (VCF). Fluorescence reports a glycine-elicited conformational change that precedes pore opening. Low concentrations of glycine, partial agonists or specific mixtures of glycine and strychnine trigger the full fluorescence signal while weakly activating the channel. Molecular dynamic simulations of a partial agonist bound-closed Cryo-EM structure show a highly dynamic nature: a marked structural flexibility at both the extracellular-transmembrane interface and the orthosteric site, generating docking properties that recapitulate VCF data. This work illuminates a progressive propagating transition towards channel opening, highlighting structural plasticity within the mechanism of action of allosteric effectors.

Pentameric ligand-gated ion channels (pLGICs), including nAChRs, 5-HT$_3$Rs, GlyRs, and GABA$_A$Rs constitute a superfamily of transmembrane receptors mediating intercellular communications in the nervous system[1,2]. They transduce the binding of a neurotransmitter at their orthosteric site into the opening of an intrinsic ion channel, leading to ion fluxes that promote cell excitation or inhibition. The orthosteric binding site also binds partial agonists, with less efficacy than full agonists to elicit channel opening, and competitive antagonists. These distinct pharmacological profiles were interpreted in terms of the concerted two-state model (Monod et al., 1965) based upon a pre-existing equilibrium between the resting and active states that is differentially shifted depending on the nature of the effector[3–5].

In between these two equilibrium states, the conformational pathway that the protein follows during activation remains elusive. Single-channel recordings of numerous mutants of the muscle nAChR analyzed by REFERs (rate equilibrium linear free energy relationships) suggested an early motion of the extracellular domain (ECD) and a late motion of the transmembrane domain (TMD) during activation[6]. Likewise, single-channel kinetics analysis on GlyRs and nAChRs were interpreted via a model involving a multistep process since the accurate fitting of the experimental data requires the introduction of intermediate states "flip"[7] or "primed"[8] in the gating transition. Although capturing structural reorganizations only indirectly, the electrophysiological data suggest that the transition is progressive, with the conformational changes starting from the ECD where the orthosteric site is located and propagating to the TMD to reach the channel gate. Moreover, computational studies of various pLGICs by Molecular Dynamics highlight their passage through a complex conformational landscape with the gating transition being composed of a progressive reorganization of the multimeric receptor architecture[9–13].

[1]Institut Pasteur, Université Paris Cité, CNRS UMR 3571, Channel-Receptors Unit, Paris, France. [2]Sorbonne Université, Collège doctoral, Paris, France. [3]Institut de Chimie de Strasbourg, UMR7177, CNRS, Université de Strasbourg, F-67083 Strasbourg, Cedex, France. [4]Sorbonne Université, 21 rue de l'École de Médecine, 75006 Paris, France. [5]Neuroscience Department, Institut Pasteur, Collège de France, Paris, France. [6]These authors contributed equally: Sophie Shi, Solène N. Lefebvre. ✉e-mail: mcecchini@unistra.fr; pierre-jean.corringer@pasteur.fr

At the structural level, fluorescence experiments provided evidence that the bacterial pLGIC homolog GLIC undergoes a progressive reorganization, with a first pre-activation step involving quaternary compaction of the ECD[14] followed by pore opening associated with a structural reorganization of the TMD and ECD-TMD interface[15]. However, direct evidence for progressive signal transduction in eukaryotic receptors is essentially lacking, despite recent cryo-EM structures of pLGICs revealing that the binding of orthosteric ligands is often accompanied by a significant reorganization of the extracellular domain (ECD), while the channel remains closed at the transmembrane domain (TMD), e.g. in 5-HT$_3$R[16,17], GABA$_A$R[18], GlyR[19], and nAChR[20]. Thus, pLGIC structures appear to be more flexible than previously anticipated, suggesting that ligands or classes of molecules could stabilize specific conformation(s) in the conformational landscape accessible to the protein. Whether the captured conformations solved by cryo-EM correspond or not to intermediates during transitions on the pathway to channel activation remains an open question. Addressing these questions requires monitoring the conformational dynamics of a fully functional receptor embedded in a plasma membrane and probing its relationship with both activation and desensitization. The voltage-clamp fluorometry (VCF) technique is perfectly suited to this purpose, since it allows the simultaneous recording of local conformational changes by fluorescence together with transmembrane current fluxes by electrophysiology.

Here, we used the α1 homomeric GlyR, which is surely among the best characterized pLGIC by electrophysiology[21,22], cryo-EM alone or in complex with ligands[16,19,23], VCF[24–31], and MD simulations[32,33]. We report the development and characterization of a mutant GlyRα1 bearing a fluorescent sensor at the interface between ECD and TMD. Its pharmacological characterization using different allosteric effectors by VCF coupled to MD simulations and docking illuminates the occurrence of intermediates, displaying dynamic properties of the ECD, on the path to channel opening consistent with a progressive mechanism of gating.

## Results

### Design of a fluorescent-quenching sensor reporting conformational changes at the ECD-TMD interface

To generate a fluorescent sensor related to the activation process, we compared the structure of the zebrafish GlyRα1 in the resting-like apo state (PDB:6PXD) with the glycine-bound open-channel structure (PDB:6PM6)[19]. Besides the well-documented contraction of the orthosteric site at the subunit–subunit interfaces in the ECD and the opening of the channel in the TMD, visual inspection also shows an expansion of the ECD-TMD interface, where the Cys-loop of each subunit moves away by nearly 3 Å from the pre-M1 region of the adjacent subunit (Fig. 1A, B). In order to monitor this reorganization, we used the tryptophan quenching technique that can sense the relative place of two residues[34,35]. This technique is particularly relevant to monitor distance changes of 5–15 Å range[14,15]. We introduced a cysteine at position Q219 in the pre-M1 region previously used for labeling with fluorescent dyes[28], together with a tryptophan in Cys-loop at position K143, in order to quench the fluorophore in a conformation-dependent manner (Fig. 1A, B).

### The quenching sensor K143W/Q219C reveals glycine-elicited fluorescence variations at much lower glycine concentrations than currents

The endogenous extracellular single cysteine of the human GlyRα1 was mutated to avoid non-specific labeling, yielding WT-C41V displaying wild-type-like properties in electrophysiology (Supplementary Fig. 1). All mutations shown here present this background C41V mutation. Upon introduction of the quenching sensor mutations, the unlabeled K143W/Q219C was analyzed by two-electrode voltage-clamp electrophysiology (TEVC). In all dose-response curves presented here, we used brief perfusion (10–30 s) of the glycine solutions to reach a current plateau with no apparent desensitization. The unlabeled K143W/Q219C displays an EC$_{50}^{current}$ that is 16-fold higher than that of WT-C41V (Supplementary Fig. 3A, B and Table 1), indicating a loss-of-function phenotype. We controlled that the mutations do not impair ion channel permeation by performing outside-out single-channel recordings (Fig. 2C) and measuring a unitary conductance of 86.7 ± 4.6 pS that is identical to that of WT-C41V (85.7 ± 4.5 pS) (Supplementary Fig. 1).

After labeling with MTS-TAMRA (Fig. 1A), K143W/Q219C was analyzed by VCF, using a dedicated recording chamber (Supplementary Fig. 2). Its EC$_{50}^{current}$ decreases by 1.6-fold, indicating that the labeling produces a slight gain-of-function that partially counterbalances the effect of the K143W/Q219C mutations. Robust glycine-elicited fluorescence variations were also observed (Fig. 2A, B). They start at non-activating glycine concentrations (5–25 μM) and reach a plateau at the beginning of the current dose-response curve, with a maximal fluorescence amplitude ΔF/F corresponding to a 12.9 ± 1.0% decrease. Therefore, the EC$_{50}^{fluo}$ is 73-fold lower than the EC$_{50}^{current}$. As a control, we also recorded the labeled Q219C/K143F and Q219C, that show weak ΔF/F (no variation and a 2 ± 0.2% decrease, respectively) (Supplementary Fig. 3B, C), indicating that the ΔF/F of the K143W/Q219C sensor is dominated by the quenching provided by the introduced tryptophan. Of note, the MTS-TAMRA used here corresponds to a commercial mixture of isomers (Fig. 1A), each of which was verified to produce the same VCF pattern separately (Supplementary Fig. 3D).

### The quenching sensor K143W/Q219C reports an "intermediate" conformational reorganization occurring before the onset of currents

Inspection of the current/fluorescence time courses and quantification by single exponential fitting reveals important features. First, the onset of both the fluorescence and current traces is faster with increased glycine concentrations whereas their offset during agonist rinsing is slower (Fig. 2D, F, Supplementary Table 3, and Supplementary Fig. 4).

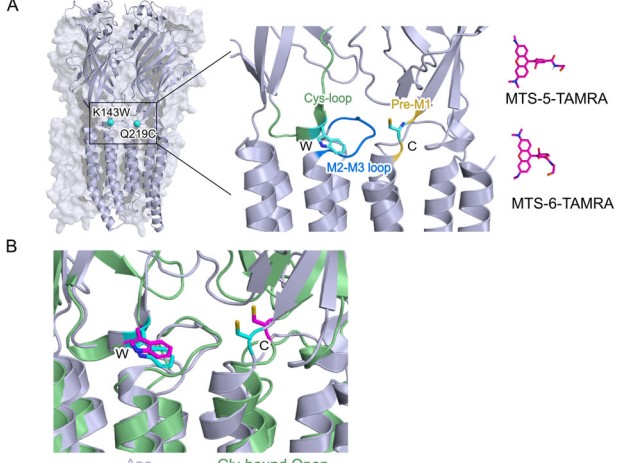

**Fig. 1 | Localization of K143W/Q219C sensor in the α1 glycine receptor structure. A** Side view of the zebrafish α1 glycine receptor structure in Apo state (PDB:6PXD) showing the position of the mutated residues K143W in Cys-loop and Q219C in Pre-M1 loop. Right panel: representation of the two isomers of 5(6)-carboxytetramethylrhodamine methanethiosulfonate (MTS-TAMRA) used for the labeling of the mutated cysteine. **B** Structural comparison of the quenching sensor K143W/Q219C between the Apo state in gray (PDB:6PXD) and Gly-bound open state in green (PBD:6PM6), residues are presented in cyan for apo state and magenta for Gly-bound open state. The distance calculated between the Cα of the two residues varied from 12.9 Å for the Apo state to 15.6 Å for the Open state.

**Table 1 | $EC_{50}$ values for current and fluorescence responses to glycine at labeled and unlabeled C41V-GlyRα1 mutants**

| | $EC_{50}^{current}$ (µM) | $n_H$ | $EC_{50}^{fluo}$ (µM) | $n_H$ | $n$ |
|---|---|---|---|---|---|
| WT without C41V MTS-TAMRA | 165.84 ± 14.06 | 1.55 ± 0.08 | | | 5 |
| WT without C41V unlabeled | 79.30 ± 11.34 | 2.16 ± 0.36 | | | 8 |
| WT MTS-TAMRA | 162.87 ± 43.21 | 1.18 ± 0.13 | | | 5 |
| WT unlabeled | 152.40 ± 28.87 | 1.33 ± 0,26 | | | 4 |
| Q219C MTS-TAMRA | 56.80 ± 6.65 | 1.84 ± 0.13 | 550.95 ± 67.55 | 1.71 ± 0.15 | 6 |
| Q219C unlabeled | 70.52 ± 3.47 | 2.23 ± 0.46 | | | 5 |
| K143W/Q219C MTS-TAMRA | 1541.12 ± 338.71 | 1.06 ± 0.11 | 21.00 ± 2.26 | 0.96 ± 0.10 | 5 |
| K143W/Q219C unlabeled | 2485.20 ± 145.08 | 1.04 ± 0.03 | | | 5 |
| K143W/Q219C/N46D/N61D MTS-TAMRA | 23029.40 ± 4281.72 | 1.41 ± 0.04 | 1428.52 ± 396.81 | 1.12 ± 0.06 | 5 |
| K143W/Q219C V280M MTS-TAMRA | 22.79 ± 11.24 | 0.544 ± 0.06 | | | 5 |
| K143W/Q219C R271Q MTS-TAMRA | | | 118.09 ± 15.11 | 0.81 ± 0.06 | 6 |
| R271C MTSR | 2863. 40 ± 766.67 | 1.36 ± 0.11 | 4517.60 ± 1335.87 | 1.12 ± 0.11 | 5 |

Second, at each glycine concentration, the rise time of the fluorescence change is systematically faster than that of the currents whereas the offset tends to be slower, particularly at glycine concentrations >200 µM. It is noteworthy that these kinetic values are limited by the rate of agonist perfusion in the recording chamber, and do not reflect the intrinsic kinetics of the receptor. For instance, upon perfusion of 200 µM Gly (Fig. 2D), the glycine concentration at the oocyte surface will progressively rise, first reaching the concentration eliciting full fluorescence variation (100 µM), then reaching after a delay in the concentration eliciting currents (200 µM). Conversely, upon glycine washing, its concentration will rapidly drop below 100 µM, causing deactivation, but will take more time to drop below the concentrations causing fluorescence variations. The relatively slow kinetics of perfusion of the agonist thus separate in time the molecular events causing the fluorescence variation and the current, the former preceding the latter.

Figure 2A, B shows that 100 µM glycine, at steady state, elicits most of the fluorescence variation but no significant current. This indicates that the receptor has completed the full movement reported by the quenching sensor with a closed channel. Hence, VCF data provide direct evidence for a structural conformational transition toward one or a cluster of conformations that are structurally distinct from both resting and active. Importantly, kinetic data show that these conformations appear before channel opening in our recording conditions (Fig. 2D, F). This suggests that they correspond to early "intermediate conformation(s)" along the allosteric transition pathway to activation that are not related to desensitization. To further investigate this possibility, we applied glycine at a high concentration (30 mM) for a prolonged period (1 min), leading to a rapid onset of the current followed by a slower decrease (34.6 ± 4.9% decrease after 1 min) caused by desensitization. The fluorescence strongly decreases during the activation phase, then reaches a plateau and remains stable during the second phase, providing evidence that the quenching sensor does not report movements related to desensitization (Fig. 2E).

**The Gly-elicited fluorescent and current changes are linked to glycine binding to the orthosteric site**

The marked separation of the current (ΔI) and fluorescence (ΔF) dose-response curves raises the possibility that both processes could be mediated by different classes of glycine binding sites within and outside the orthosteric site. To challenge this possibility, we introduced two mutations into the orthosteric binding site (N61D/N46D, Fig. 3A) that strongly decrease the affinity for glycine[36]. The K143W/Q219C/N61D/N46D shows a parallel rightward shift of both ΔI and ΔF curves (15-fold and 68-fold increase in $EC_{50}^{current}$ and $EC_{50}^{fluo}$, respectively) (Fig. 3B, C and Table 1), indicating that both processes are strongly linked to glycine binding to the orthosteric site. We also found that ΔF

changes are sensitive to known allosteric hyperekplexia mutations at the ECD-TMD interface. Combined with K143W/Q219C, the loss of function R271Q[37], located at the top of the M2 helix, fails to generate currents (certainly due to the combination with K143W) but a ΔF is observed and its curve is shifted to higher glycine concentrations (Supplementary Fig. 5A). Conversely, combined with K143W/Q219C, the gain of function V280M[38], located within the loop linking the M2 and M3 helices (M2-M3 loop) does not show fluorescence variation, but currents with a ΔI curve shifted to lower concentrations. In addition, it shows high leak currents in the absence of glycine that are inhibited by strychnine, indicating constitutive channel openings (Supplementary Fig. 5B, C).

**Partial agonists, strychnine, and propofol differentially affect the current and fluorescence variations**

First, we investigated the effect of agonists that bind at the orthosteric site but are less effective than glycine at activating the receptor, i.e., partial agonists. We selected β-alanine and taurine. They trigger respectively 54–98% and 20–42% of the glycine-elicited maximum current[39–41] in TEVC[39,41–43]. On K143W/Q219C, we found that their efficacy decreases, eliciting only 6.1 ± 2.2% and 2.0 ± 1.4% of the glycine-elicited maximum current respectively (Fig. 4A, B). Such a decrease in efficacy is commonly observed for loss-of-function mutants[31]. Surprisingly, both partial agonists elicit the full fluorescence variation recorded with glycine. β-Alanine and taurine show a large difference in the ΔF and ΔI dose-response curve, their $EC_{50}^{fluo}$ being respectively 66- and 70-fold lower than their $EC_{50}^{current}$ (Table 1). β-Alanine and taurine thus trigger the full transition mediating the fluorescence change, but they are weakly effective to trigger the "downward" process of channel opening.

Next, we investigated the competitive antagonist strychnine. When strychnine is applied alone at 5 µM, no current is observed as expected, but the fluorescence increases over the baseline in a direction opposite to the glycine-elicited quenching (dequenching). It then reaches a plateau corresponding to 38 ± 6% in the absolute amplitude of the maximal glycine-elicited fluorescence variation. We then applied strychnine (5 µM) during the perfusion of glycine at different concentrations (Fig. 5). At 100 µM glycine, strychnine inhibits both the current and fluorescence variations elicited by glycine. As before, strychnine tends to increase the fluorescence over the baseline, but the effect is not significant. At 1 mM of glycine, strychnine still totally inhibits the current, but surprisingly the glycine-elicited fluorescence quenching is decreased by only 48 ± 4%. Finally, at 10 mM of glycine, strychnine inhibits only 77 ± 2% of the current, while the fluorescence is almost not affected. Therefore, the co-application of strychnine with various concentrations of glycine differentially affects the fluorescence variations and currents.

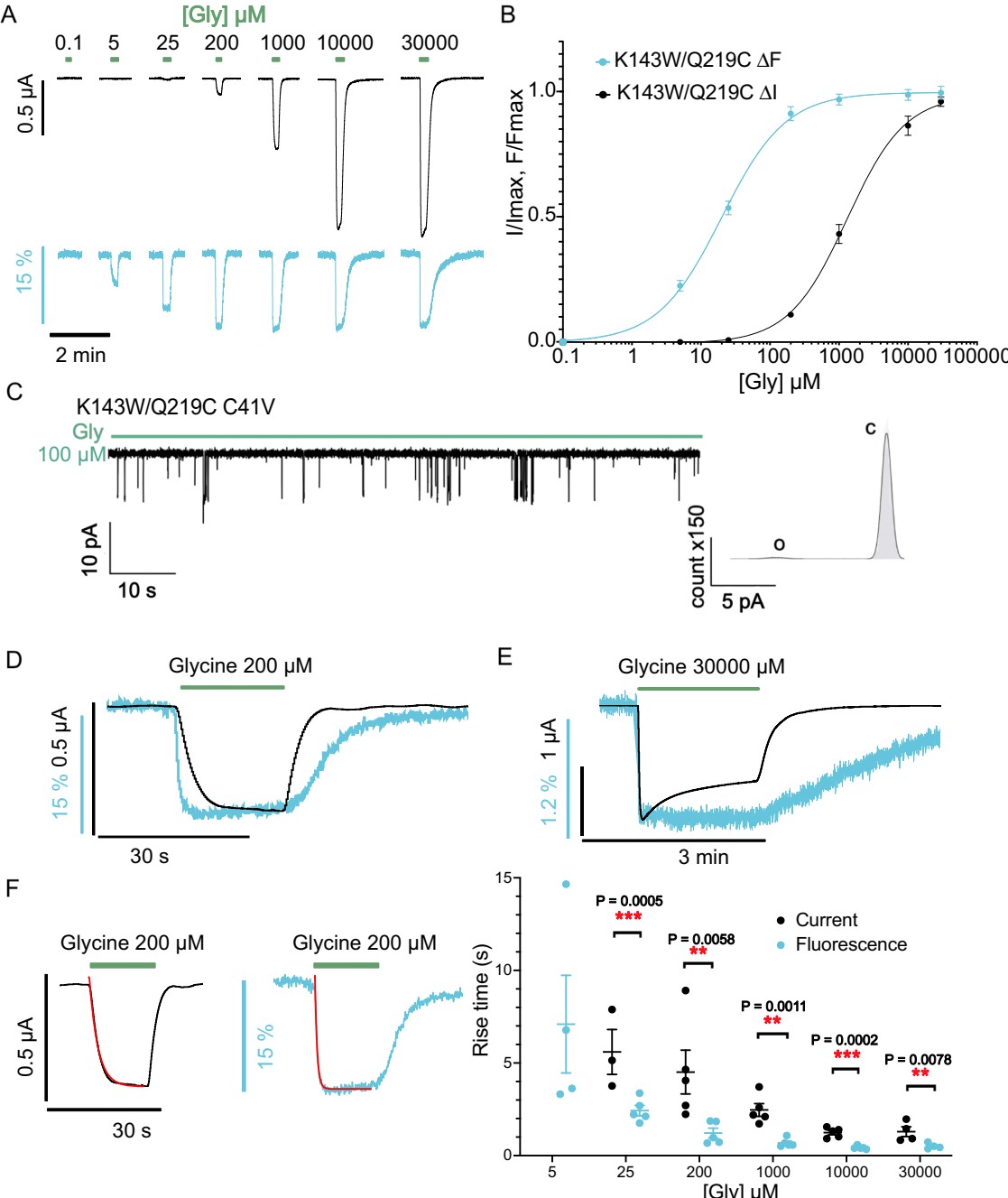

**Fig. 2 | Electrophysiological and fluorescence characterization of K143W/Q219C sensor by VCF on C41V GlyR. A** Representative VCF recordings on oocytes of the mutant labeled with MTS-TAMRA (black for current and cyan for fluorescence). Glycine application triggers a current variation and a fluorescence quenching phenomenon with a maximum fluorescence variation that reaches 12.1 ± 1.1 % of ΔF/F. At low concentrations of glycine (under 25 μM), only fluorescence quenching is observed without any current. **B** ΔI (black) and ΔF (cyan) dose-response curves with mean ± SEM values (*n* = 5). **C** Left panel: representative trace of single-channel recordings obtained in outside-out configuration recorded at −100 mV with concentrations of glycine at 100 μM. Right panel: histograms of current amplitude representing the closed state (c) and the open state (o). **D** Superimposition of the current and fluorescence recordings evoked by 200 μM

glycine application shows that the onset of the fluorescence is faster than the onset of the current and that the fluorescence offset is slower than the current offset. **E** Representative recording with a high glycine concentration (30 mM) application shows that the desensitization triggered by a prolonged glycine application does not impact the fluorescence variation. A mean decrease of 30.63 ± 4.90% of the current elicited by 30 mM of glycine is observed (*n* = 4). **F** Left panel: examples of single exponential fitting (red line) of the current (left) and fluorescence (right) traces onset. Right panel: time constants τ (onset) values obtained via single exponential fitting with mean and SEM error bars at different glycine concentrations (*n* = 3 and 4 for glycine concentrations under 25 μM and *n* = 5 for other concentrations). Unpaired two-sided student *t*-test indicates the significance of the difference between fluorescence and current onset (**P < 0.005; ***P < 0.0005).

Last, we investigated the general anesthetic propofol, a positive allosteric modulator of GlyRα1 that binds in the TMD of pLGICs[44–47]. At 200 and 300 μM, propofol alone elicits a fluorescence variation of about 3–5% ΔF/F without activating the receptor, while at higher

concentrations it also activates the currents in the absence of glycine (Supplementary Fig. 6). At a lower 100 μM concentration, propofol alone does not elicit significant ΔF. Interestingly, the co-application of propofol with a range of glycine concentrations shows that it

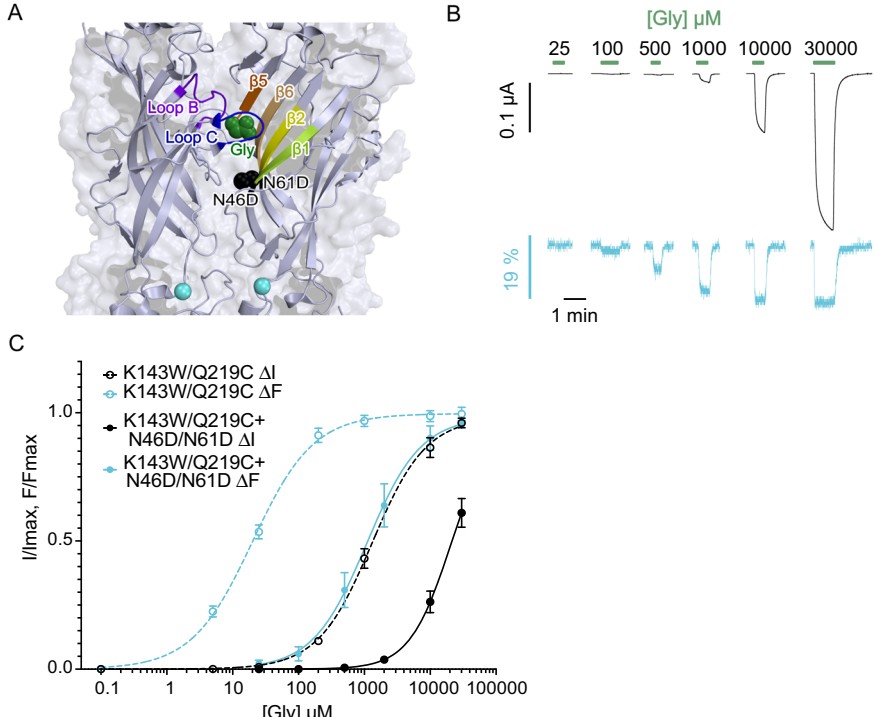

**Fig. 3 | Effect of N46D/N61D mutations on ΔI and ΔF curves of K143W/Q219C on C41V GlyR. A** Side view of zebrafish α1 glycine receptor structure in Apo state (PDB:6PXD) with details of loops and β-sheets forming the orthosteric site and the localization of N46D/N61D mutations. The glycine molecule is represented in green and N46D/N61D mutated residues (spheres) in black. **B** Representative VCF recordings of the mutant K143W/Q219C + N46D/N61D (current in black and fluorescence in cyan). **C** ΔI (cyan) and ΔF (black) dose-response curves with mean ± SEM show a parallel shift of the current and fluorescence curves of the K143W/Q219C/N46D/N64D mutant (solid line; n = 5) compared to K143W/Q219C alone (dotted line).

potentiates by 21 ± 7% the maximal currents and shifts by 8-fold the ΔI curve to lower concentrations. In contrast, propofol does not affect significantly the ΔF curve (Fig. 4C and Supplementary Table 1). Propofol thus does not alter the transition mediating the fluorescence change, but specifically facilitates the "downward" process of channel opening.

In conclusion, various pharmacological conditions yield a common phenotype with a change in ΔF with comparatively weaker or no change in ΔI: they are low glycine concentrations, partial agonists, specific mixtures of glycine and strychnine and 200–400 μM of propofol alone. This phenotype suggests that a substantial fraction of the receptors has underwent the local conformational motion causing quenching of K143W/Q219C in the lower part of the ECD but with a closed channel. However, it is possible that the glycine-strychnine combination and propofol alone change the environment of the sensor in a similar way as partial agonists and agonists do, giving similar ΔF signature, while the rest of the ECD could present different conformations. This is especially possible for propofol alone as it binds to the TMD of GlyRs and might thus present an alternative mechanism.

### The R271C mutant also shows a relatively increased fluorescence variation when activated by taurine as compared to glycine
Among pLGICs, the GlyR has been extensively studied by VCF by the team of Lynch et al., with the labeling of the upper M2 helix at position R271[25,31], the orthosteric site, the loop 2 and pre-M1 regions[28,30], and the TMD[26,27]. In all cases, the fluorescence dose-response curves were superimposed or right-shifted as compared to the current dose-response curve under glycine application (Supplementary Fig. 7). Thus, the leftward shift of the ΔF dose-response curve shown here seems unique.

However, another landmark of K143W/Q219C concerns the action of partial agonists, that produce much higher maximal ΔF/ΔI ratios

than glycine. Interestingly, R271C[31] was also reported to display such a phenotype with taurine. On our VCF setup, with glycine, R271C produces an 18-fold shift of the ΔI curve to higher concentrations as compared to WT, and a near superimposition of the ΔI and ΔF curves. Taurine at saturation elicits significantly higher ΔF than ΔI, respectively 21 ± 3.4% and 5.4 ± 2.3% of glycine values. In addition, for taurine, the ΔF curve is shifted by 2-fold to lower concentrations as compared to the ΔI curve (Fig. 6A, B). These data, that are in qualitative agreement with those of Pless et al. (2007), indicate that a fraction of the taurine-elicited states of R271C presents a fluorescence variation with a closed channel.

### Molecular dynamics simulations of a cryo-EM partial-agonist bound state recapitulate the pharmacological profile of the intermediate state(s)
A recent cryo-EM analysis of the zebrafish GlyR[19] has revealed that the partial agonists taurine and GABA may stabilize a previously unseen closed-channel state (referred to as "tau-closed") characterized by a significant reorganization of the ECD compared to the apo state. We hypothesized that the intermediate conformation(s) illuminated by VCF is structurally similar to tau-closed. To explore this hypothesis, we carried out all-atom MD simulations of the zebrafish apo-closed (PDB:6PXD), tau-closed (PDB:6PM3), and tau-open (PDB:6PM2) structures embedded in a POPC bilayer. In addition, since strychnine nicely fits the orthosteric site of apo-closed, we performed MD simulations of the strychnine-closed (stry-closed) complex as well. For each system, five independent simulations of 100 ns were carried out.

The structural stability of the protein was analyzed by monitoring the root-mean-square fluctuations (RMSF) from the average structure extracted from MD (Fig. 7A). In addition, we calculated the evolution of the twisting and blooming angles of the ECD during each simulation (Supplementary Fig. 8 and Supplementary Table 4). The simulations

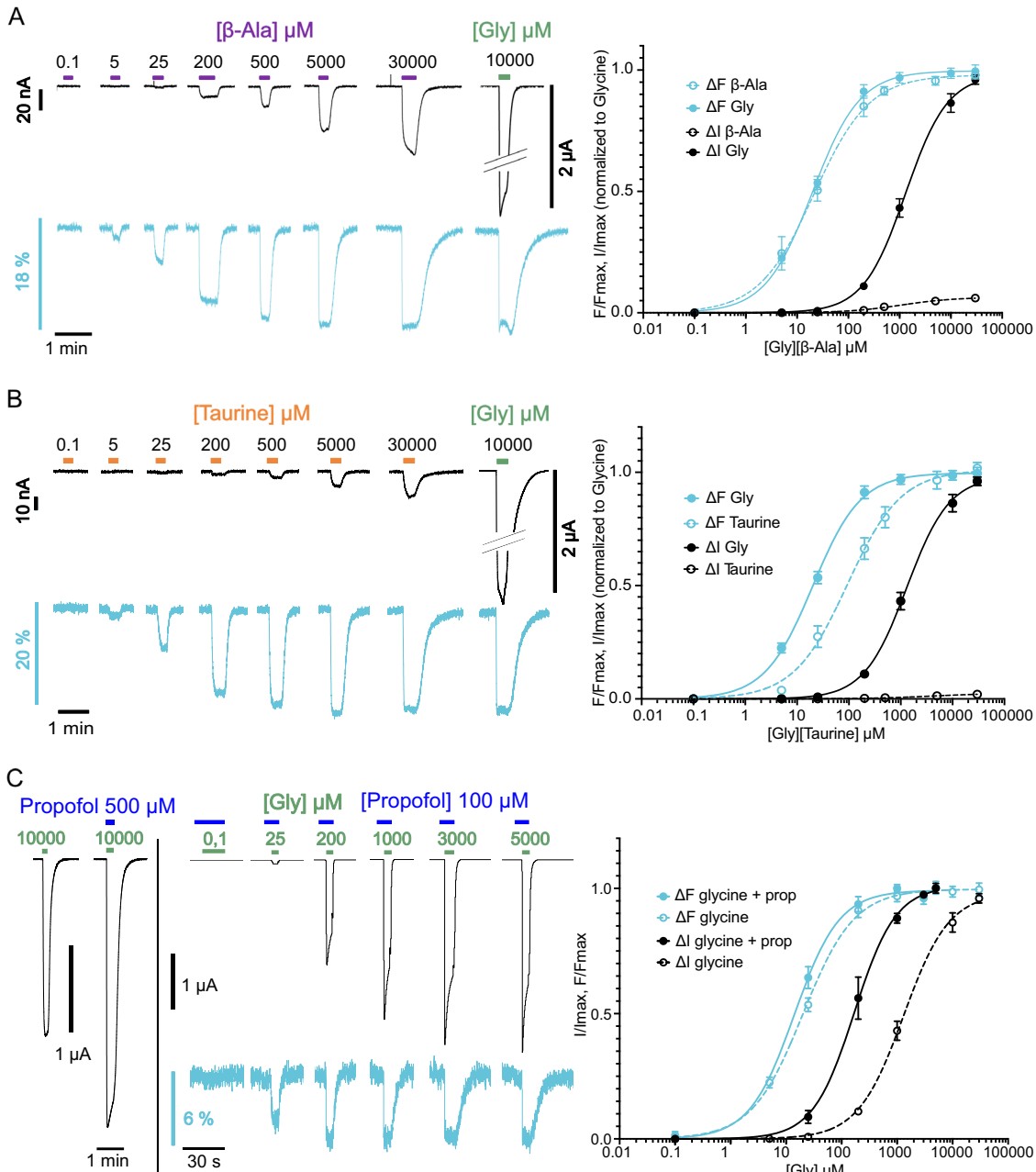

**Fig. 4 | Electrophysiological and fluorescence characterization of partial-agonist (β-alanine and taurine) and propofol effects on K143W/Q219C sensor on C41V GlyR. A** Right panel: Representative VCF recordings on an oocyte challenged with both β-alanine and glycine. Left panel: representative VCF recording on oocytes under the β-alanine application. Right panels: the ΔI (cyan) and ΔF (black) curves (normalized to the glycine maximal response recorded on each individual oocyte) with mean and SEM for K143W/Q219C under glycine (solid line) and β-alanine (dotted line) application. The ΔF curves for both molecules are superimposed but the efficacy of β-alanine to activate the receptor is lower than that of glycine (6.1 ± 2.2% of glycine maximum response in current, *n* = 6). **B** Right panel: Representative VCF recordings on an oocyte challenged with both taurine and glycine. Left panel: representative VCF recording in oocytes under the taurine application. Right panel: ΔI (cyan) and ΔF (black) curves (normalized to the glycine maximal response recorded on each individual oocyte) with mean and SEM for K143W/Q219C under glycine (solid line) and taurine (dotted line) application. Taurine elicits only 2.0 ± 1.4% of glycine maximum current, *n* = 6. **C** Left panels: Representative VCF recording on an oocyte challenged with maximal glycine concentration in the presence and absence of propofol at 500 μM. Middle panel: representative VCF recording on an oocyte under glycine and propofol co-application. Right panel: ΔI (cyan) and ΔF (black) dose-response curves with mean and SEM for K143W/Q219C under glycine and propofol (solid line, *n* = 6) and glycine alone (dotted line) application.

show that: i. apo-closed is the most flexible conformational state of the receptor (Fig. 7A) and strychnine binding stabilizes the bloomed conformation (expended ECD as seen in the Apo-closed cryo-EM structure), rigidifying the ECD; ii. the ECD of tau-closed is remarkably more flexible than that in tau-open despite the apparent structural similarity in the cryo-EM coordinates that show an un-bloomed (compaction of the ECDs) structure; iii. tau-closed has a weaker

affinity for taurine than tau-open as evidenced by four spontaneous unbinding events in the tau-closed simulations, that are associated with significant blooming of the structure (Supplementary Fig. 10); iv. the C loop that is critical for orthosteric-ligand binding is more flexible in tau-closed (RMSF of 3.5 Å) than in tau-open (RMSF of 2.5 Å) or stry-closed (RMSF of 2.0 Å); v. the conformational dynamics of the ECD-TMD interface (i.e., β1–β2 loop, Cys-loop, and the β8–β9 loop) is

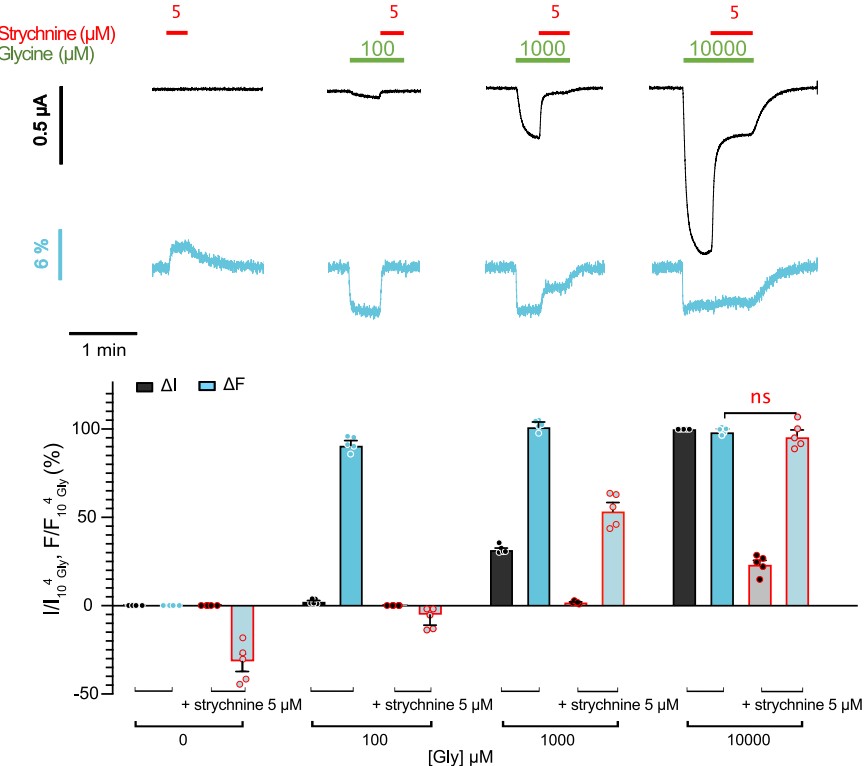

**Fig. 5 | Differential inhibition of strychnine on fluorescence and current on K143W/Q219C sensor on C41V GlyR.** Upper left panel: Representative VCF recording where the application of strychnine alone at 5 μM triggers a fluorescence dequenching. Upper right panels: glycine is applied at different concentrations first alone (green bar) and then in a mixture with 5 μM strychnine (red bar plus green bar), showing differential strychnine-elicited inhibition of current and fluorescence depending on the glycine concentration ($n = 5$). Lower panel: fluorescence (cyan) and current (black) variations normalized to the fluorescence and current variations under 10 mM of glycine (mean and SEM ($n = 5$)). "ns" denotes that the ΔF (under glycine) and ΔF (under strychnine inhibition) are not significantly different; $P < 0.05$ (Unpaired two-sided student $t$-test).

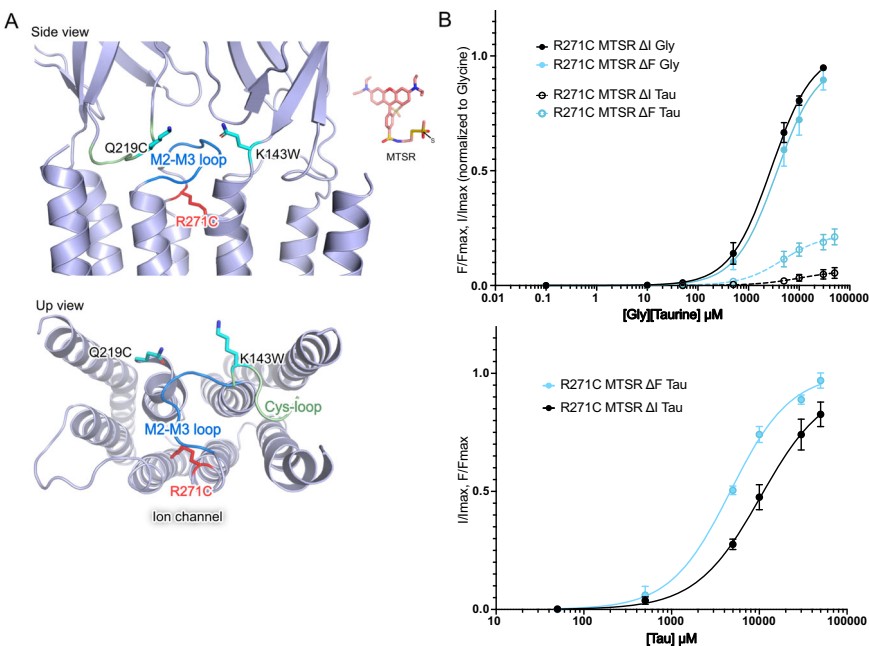

**Fig. 6 | Characterization of R271C by VCF on C41V GlyR. A** Localization of the R271C mutation on the GlyR compared to the sensor K143W/Q219C (PDB:6PXD). **B** Upper panel: ΔI (cyan) and ΔF (black) dose-response curves (normalized to the glycine maximal response recorded on each individual oocyte) with mean and SEM error bars for R271C under glycine (solid line) and taurine (dotted line) application. Taurine elicits only $5.4 \pm 2.3\%$ of glycine maximum current, $n = 5$. Lower panel: ΔI (cyan) and ΔF (black) dose-response curves normalized to the maximum value with mean and SEM error bars for R271C under the taurine application.

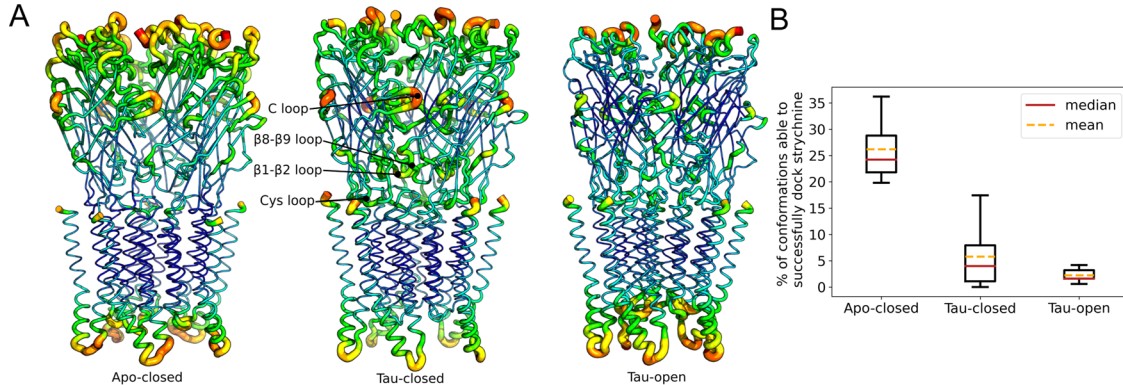

**Fig. 7 | Dynamic "personality" of the tau-closed state. A** Root-mean-square fluctuations (RMSF) from the average structure extracted from 0.5 μs MD simulations in explicit solvent/membrane are shown for three conformational states of GlyR captured by cryo-EM. Apo-closed corresponds to the resting state and tau-open to the active state of the receptor. For each state, RMSF values ranging from 0.5 to 4.5 Å are shown by the color (from blue to red) and the thickness (from thin to thick) of the sausage representation (**B**). State-based docking of strychnine. The boxplot defines the center at the median (red line) and mean (yellow dashed line),

minima and maxima are indicated by whiskers, and the box represents data between the first and third quartiles. Relative strychnine-binding affinities for apo-closed, tau-closed, and tau-open was probed by docking strychnine to an ensemble of 500 protein snapshots sampled by MD and comparing the success rate of docking; i.e., a docking experiment was considered as successful if the docking score was within 10% of the score obtained in the strychnine-bound X-ray structure (stry-closed). The mean (yellow) or median (red) dashed lines show that strychnine binding in tau-closed is significantly stronger than in tau-open.

enhanced in tau-closed relative to apo-closed and tau-open. Taken together, the simulation results highlight that the tau-closed conformation features a surprising dynamic character, which is remarkably different from that of tau-open and apo-closed and could not be deduced from the comparison of static cryo-EM structures only.

To explore the correlation between MD and VCF data, we quantified the reorganization of the fluorescence quenching sensor, as well as the binding of strychnine.

First, we calculated the average distance between the Cα carbons of the VCF quenching sensor (corresponding to K143W/Q219C). It is 13.0 ± 0.7 Å in stry-closed and apo-closed and this value increases to 14.7 ± 1.4 Å and 16.4 ± 1.0 Å in tau-closed and tau-open, respectively. Therefore, the significant reorganization seen in the simulations of the sensor from the apo to the tau-closed conformation is compatible with a change in fluorescence during this transition. We speculate that, during the separation of the Cα carbons, the rather long and flexible TAMRA "side chain" anchored at C219 reorganizes to interact more frequently with W143, generating fluorescence quenching.

Second, the binding of strychnine to the various conformations of GlyR was explored by docking. As expected, favorable docking scores were obtained in stry-closed (−12.1 kcal/mol) and apo-closed (−10.3 kcal/mol) from cryo-EM. By contrast, both tau-open and tau-closed from cryo-EM feature an orthosteric site that is too small to accommodate strychnine and the docking scores were largely unfavorable. To account for the intrinsic flexibility of the protein, the docking experiments were repeated using protein conformations extracted from the MD trajectories. Assuming that a docking experiment was successful when the docking score was <−9.3 kcal/mol (i.e., within 10% of the score obtained in stry-closed), the success rate for docking strychnine was determined by counting the fraction of successful docking over 500 protein snapshots from MD of apo-closed, tau-closed, and tau-open. As shown in Fig. 7B, docking of strychnine was successful at 26% (std dev 6.5%) in apo-closed, 6.6% (std dev 7.0%), and with large deviations among replicas in tau-closed, and only 2.2% (std dev 1.4%) in tau-open. We conclude that the enhanced flexibility of the tau-closed structure makes it very different from tau-open and compatible with strychnine binding.

Although this analysis was carried out on the WT zebrafish receptor, we performed control MD simulations of the K143W/Q219C/C41V triple mutant in the tau-open and tau-closed states, giving similar results (Supplementary Fig. 9). The simulation results support the conclusion that the intermediate conformation(s) revealed by VCF, concerning both (partial) agonists and glycine-strychnine combinations, is consistent with a highly dynamic cluster of states as described by MD simulation started from the cryo-EM tau-closed structure.

## Discussion

The VCF data on K143W/Q219C provide structural evidence for an early intermediate step in the global activation path of GlyR. Indeed, upon glycine application, they reveal the emergence of one or more intermediate conformation(s) characterized by a change in fluorescence but no current, followed by channel opening with no further change in fluorescence. Two lines of evidence support that the intermediate conformation(s) illuminated by VCF is on-path to the global activation process: (i) Receptors in this intermediate conformation(s) are activatable since they appear before the onset of the currents, and are thus not desensitized, (ii) Receptors in the intermediate conformation(s) are stabilized by partial agonists, low glycine concentrations, and mixtures of glycine and strychnine, showing a pharmacological profile "in-between" those stabilizing the resting and active states.

It is noteworthy that the mutations introduced here to anchor the fluorescence quenching sensor (K143W/Q219C) cause an 8-fold increase in EC$_{50}$$^{current}$ of glycine, and a decreased efficacy of the partial agonists to activate the receptor, yet with an intact conductance of the channel. This indicates a destabilization of the active state. Concerning fluorescence, the ΔF curve displays higher sensitivity than the ΔI curve, indicating that partial binding of glycine is sufficient to trigger the resting to intermediate(s) conformational transition. In addition, strychnine alone produces a fluorescence variation in the opposite direction to agonists, suggesting that the intermediate(s) conformation(s) are already substantially populated in the absence of ligand, with strychnine shifting the spontaneous equilibrium back toward the resting state[48]. For partial agonists, they even mainly stabilize the intermediate(s) conformation at saturation, with a very minor representation of the active state.

Interestingly, we found that a non-activating concentration of propofol potentiates the currents without change of the ΔF curve, suggesting a specific action of propofol on the intermediate(s) to active transition. We also report that the gain of function mutant V280M does not show fluorescence changes. We suggest that this mutant could populate mainly the intermediate(s) state in the absence of an agonist.

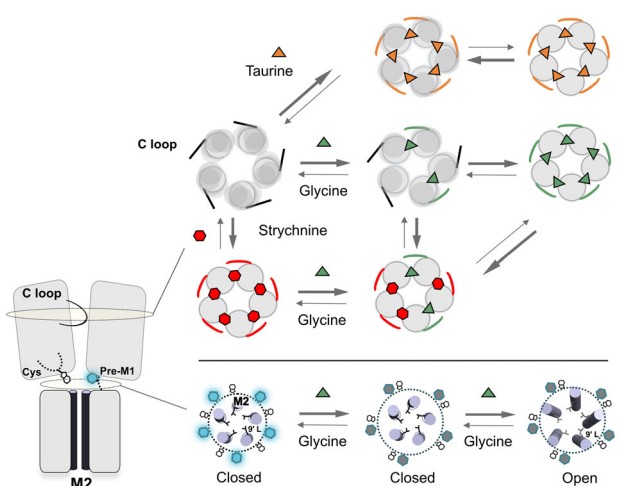

**Fig. 8 | Hypothetical transition pathway of the GlyR.** Left panel: side-view cartoon representation of two subunits of the GlyR. C loop contributing to the orthosteric site, the quenching sensor, the TAMRA (hexagon), and tryptophane (indole) are represented. Upper panel: top view representation of the receptor in different conformations with the ECD of each subunit shown in a circle and C loop as a line. Lower panel: top view representation of the TMD showing the M2 helices and the L9' that closes the pore in the resting and intermediate(s) conformations. The black circle represents positions where the fluorescent-quenching sensor is grafted. The indole goes closer to the TAMRA from the resting to the intermediate(s) conformations, generating a quenching of fluorescence.

An important observation from VCF is that taurine binding stabilizes primarily the intermediate state(s). The structure of the "tau-closed" state of the zebrafish GlyR recently solved by cryo-EM is a strong candidate for an early on-pathway intermediate during the global activation process[19]. However, docking experiments show that this structure is unable to accommodate the bulky antagonist strychnine, a feature that is inconsistent with our VCF data. Interestingly, MD simulations of the tau-closed structure provide a way to explain this discrepancy. In fact, the simulations reveal that the ECD is surprisingly dynamic in the tau-closed state and samples the spontaneous opening of the orthosteric site on the sub-μs time scale, which makes it compatible with strychnine binding. Combined with the VCF data, these observations suggest that receptors in the presence of mixtures of strychnine and agonists would keep an overall intermediate-like conformation with significant asymmetry at the ECD and at the orthosteric sites. Some sites would bind agonists/partial agonists in an active-like compact state, while others would bind strychnine in a resting-like expanded state. We speculate that the same mechanism applies in the case of partial occupation of the orthosteric sites by glycine, accounting for a maximal fluorescence variation at low glycine concentration with little current, which explains the large separation of the ΔI and ΔF dose-response curves.

The simulations reveal that the tau-closed state presents a unique dynamic personality that is remarkably different from those at the resting and active states. In fact, a comparison of the atomic fluctuations in the different conformations reveals that the flexibility of the ECD-TMD interface is surprisingly enhanced in tau-closed relative to both apo-closed (resting-like) and tau-open (active-like). We speculate that the unique dynamic signature of tau-closed originates from the hybrid nature of this state, which features an ECD in a nearly active conformation along with a resting-like TMD endowed with a closed channel. This apparent mismatch at the ECD-TMD interface, which enhances the flexibility of the ECD domain and its blooming in tau-closed, favors the structural plasticity for promiscuous orthosteric-ligand binding. This hypothesis is consistent with the large reorganization of the pre-M1 and Cys-loop regions probed by VCF in the absence of currents and suggests that the tau-closed structure and its

dynamic character provide a reasonable representation of the early intermediate conformations found not only with partial agonists, but also with glycine at sub-saturation or with glycine and strychnine mixtures.

From combined VCF and MD data, we thus propose a speculative model of receptor activation depicted in Fig. 8, where the receptor transits through a cluster of highly dynamic intermediate conformations, allowing significant local asymmetry at the level of the orthosteric site (and also potentially in the upper ECD), but a more symmetric organization at the ECD-TMD interface where the fluorescent sensors are anchored. Although we cannot exclude the existence of alternative intermediate states in different pharmacological conditions e.g. involving changes in the local environment of the fluorescence sensors, the MD simulations of the cryo-EM Tau-closed state provide a reasonable and unifying explanation of the intermediate states recorded with (partial) agonists and the glycine-strychnine mixture.

Our data also suggest that a fraction of R271C receptors, when activated by taurine, display an intermediate-like conformation. It is noteworthy that R271C and K143W/Q219C cause a similar 10-fold increase in $EC_{50}^{current}$ of glycine (Supplementary Fig. 7). We thus speculate that their major difference relies on the resting to intermediate equilibrium, which is strongly shifted toward the intermediate(s) in K143W/Q219C and much less in R271C. Interestingly, K143W/Q219C and R271C are located at both extremities of the M2-M3 loop which undergoes a large outward movement during gating (Fig. 6A).

In regards to the WT zebrafish GlyR studied by cryo-EM, literature data suggest a contribution of around 7% of Tau-closed states at taurine saturation[19]. It is thus likely that, in the WT human GlyR context, the intermediate(s) state would also be sparsely populated in the presence of partial agonists. We speculate that the K143W mutation, strongly stabilizes the intermediate(s) conformation, thus allowing its robust monitoring by VCF. In retrospect, this might explain why the "intermediate" phenotype has rarely been observed in the previous VCF studies of the GlyR. Of note, previous single-channel works on nAChRs and GlyRs suggested the contribution of flipped/primed intermediate states on receptor activation. Flipped/primed are incompletely stabilized by partial agonists, underlying their limited efficacy to fully activate the receptor[7,8]. This is in contrast with the intermediate conformation(s) unraveled here, which is fully stabilized by partial agonists and consequently must occur earlier than the flip/prime transition.

Phenotypes resembling the one presented here have been already described for the bacterial channel GLIC and cationic pLGICs. With GLIC, we reported a "pre-activation" phenotype for a series of quenching sensors at the ECD interface and M2-M3 loop[14,15]. They all report fluorescence variations at non-activating agonist (proton) concentrations and the fluorescence variation was faster (ms range measured by stopped-flow) than the onset of activation (10–100 of ms measured by patch-clamp)[49]. It is noteworthy that the M2-M3 loop is located just in between the quencher (W143) and the fluorophore (C219) in the GlyR structure (Fig. 1).

In the case of cationic pLGICs using VCF, labeling of the muscle nAChR (αγδβ subtype) at the top of TMD (position 19' located at the upper end of M2 helix and facing the ECD) reports a 100-fold lower $EC_{50}^{fluo}$ as compared to $EC_{50}^{current}$ for ACh (binding the αδ interface), while no difference was observed for epibatidine (binding the αγ interface). Moreover, the change in fluorescence occurs with fast kinetics and pharmacological considerations support the occurrence of a conformational transition toward a singly liganded closed-channel state[50]. Labeling at the 19' position has also been performed on the 5-HT$_3$R, consistent with monitoring a pre-active reorganization[17]. Finally, labeling of the α4β4 receptor in the extracellular loop 5 of α4 shows a fivefold lower $EC_{50}^{fluo}$ as compared to $EC_{50}^{current}$ for ACh suggesting that loop 5 moves before channel activation. Additionally, the antagonist DHβE elicits also a robust fluorescent change[51] which

suggests a highly dynamic nature of loop 5 in the activation and inhibition process. Altogether, data on the GlyR, GLIC, and possibly nAChR/5HT₃R provide evidence for intermediate transitions involving structural reorganizations mainly at the ECD and/or ECD-TMD interface. This idea is in line with REFERs analysis of the muscle nAChR, that suggests that the orthosteric site and M2-M3 loop move very early during the activation transition[6]. It is also consistent with simulations describing the gating isomerization in terms of a combination of two consecutive quaternary transitions named twisting and blooming[2]. In this view, the intermediate conformation(s) illuminated by VCF would correspond to the completion of the un-blooming isomerization (i.e., compaction of the ECD) in a still closed-channel receptor. The apparent absence of significant twisting during GlyR activation (Fig. S7) and its role in gating remains to be understood. Altogether, while the detailed activation mechanism may differ between different pLGIC subtypes, the collected data highlight a general activation mechanism in pLGICs characterized by progressive conformational propagation from the ECD to the TMD.

In conclusion, this work illustrates the interest in combining VCF and MD approaches to identify functional intermediate conformations, characterize their flexibility, and annotate Cryo-EM structures to functional states. Noteworthy, such intermediate conformations may contribute to signal transduction in the postsynaptic membrane. The progressive transition also illuminates the mode of action of allosteric effectors and may be valuable for drug-design purposes.

## Methods

### Site-directed mutagenesis
The gene coding the full-length human glycine α1-subunit was cloned into the pMT3 plasmid containing an Ampicillin resistance gene. The PCR reaction was done with CloneAmp Hifi premix from Takara.

### α1 homomeric glycine receptor expression in oocytes
*Xenopus Laevis* oocytes at stage VI are ordered from Portsmouth European Xenopus resource center and Ecocyte Biosciences and kept in Barth's solution (87.34 mM NaCl, 1 mM KCl, 0.66 mM CaNO₃, 0.72 mM CaCl₂, 0.82 mM MgSO₄; 2.4 mM NaHCO₃, 10 mM HEPES, and pH adjusted at 7.6 with NaOH). Ovary fragments obtained from Portsmouth European Xenopus resource center are treated as previously described[52]. cDNA coding the α1-subunit at 80 ng/μL is co-injected with a cDNA coding for GFP at 25 ng/μL into the oocyte nucleus by air injection. The injection pipettes are done with the capillary pipette (Nanoject II, Drummond) and PC-10 dual-stage glass micropipette puller. Injected oocytes are incubated at 18 °C for 3–4 days for expression.

### Labeling of the cysteine
MTS-TAMRA and MTSR (Cliniciences) are dissolved in DMSO to obtain a final stock concentration of 10 mM and conserved at −20 °C. For labeling, oocytes expressing the receptors are incubated in 10 μM MTS-TAMRA (diluted in Barth's solution to obtain a final DMSO concentration of 0.1%) at 16–18 °C for 20 min before the recording. The procedure is similar for other fluorescent dyes (MTS-5-TAMRA, MTS-6-TAMRA). For MTSR, oocytes are incubated in 10 μM MTSR at 16–18 °C for less than 1 min.

### Two-electrode voltage-clamp on oocytes
Oocytes expressing GlyR constructs are recorded in ND96 solution (96 mM NaCl, 2 mM KCl, 5 mM HEPES, 1 mM MgCl₂, 1.8 mM CaCl₂, and pH adjusted at 7.6 with NaOH). Molecules that are not fully soluble in an aqueous solution are first dissolved in DMSO and then diluted in ND96 to have a DMSO final concentration <0.1%. Currents were recorded by a Warner OC-725C amplifier and digitized by Digidata 1550 A and Clampex 10 (Molecular Devices). Currents were sampled at 500 Hz and filtered at 100 Hz. The voltage-clamp is maintained at

−60 mV at room temperature. Recording pipettes were made with Borosilicate glass with filament (BF150-110-7.5, WPI) with a PC-10 dual-stage glass micropipette puller to obtain pipettes of resistances comprised between 0.2 and 2 MΩ.

### Voltage-clamp fluorometry on oocytes
Injected oocytes are placed in a home-designed chamber with the animal pole turning toward the microscope objective. Currents were recorded by GeneClamp 500 voltage patch-clamp amplifier (Axon Instruments) and digitized by Axon Digidata 1400 A digitizer and Clampex 10.6 software (Molecular Devices). TAMRA excitation is done by illumination with a 550 nm LED (pE-4000 CoolLED, 15–20% intensity) through an FF01-543_22 bandpass filter (Semrock). The fluorescence emission goes through an FF01-593_40 bandpass filter (Semrock) and is collected by a photo-multiplicator H10722 series (HAMAMATSU). Recording pipettes were the same as those prepared for TEVC. Currents were sampled at 2 kHz and filtered at 500 Hz. The voltage-clamp is maintained at −60 mV and at room temperature. The recording chamber has been designed to perfuse only the part of the oocyte from which the fluorescence emission is simultaneously collected (Fig. S3). This chamber has several advantages over chambers that allow to record the current of the entire membrane of the oocytes: (1) approximately the same population of receptors is recorded in current and fluorescence simultaneously, (2) the chamber allows for overexpression of the receptor on the oocyte membrane to increase the fluorescence signal without generating huge currents that make the clamp impossible to maintain during a prolonged period. The oocyte is placed in a hole and the geometry of the perfusion channels has been designed to "suck" the oocyte and stick it to the hole through a Venturi effect, ensuring efficient sealing between the two compartments. A limitation of the chamber is that the design is compatible with only relatively slow perfusion of the agonist, with rapid perfusion often expelling the oocyte out of the hole.

### Voltage-clamp fluorometry data analysis
For each construct, we performed the experiments at least on two different batches of oocytes (oocytes from different ovary fragments and different animals) to obtain at least $n = 5$. Data were analyzed by Clampfit software (Molecular Devices) and are filtered by a boxcar filter. The baseline was corrected from leaking currents and measurements were done to the peak of each response. Dose-response curves were fitted to the Hill equation by Prism GraphPad software, and the error bars represent the SEM values. Rise time and decaytime analysis were calculated in Clampfit by monoexponential fits of the onset and the offset of the recordings. Statistical analyses were done using a Student *t*-test in Prism Graphpad.

### Expression in cultured cells
Human Embryonic Kidney 293 (HEK-293, ATCC CRL-1573) cells were cultured in Dulbecco's modified Eagle's medium (DMEM) with 10% FBS (Invitrogen) in an incubator at 37 °C and 5% CO₂. After being PBS washed, trypsin-treated (Trypsine-EDTA; ThermoFisher Scientific), and seeded on Petri dishes, cells were transiently transfected using calcium phosphate-DNA co-precipitation with glycine receptor constructs (2 μg DNA) and a construct coding for a green fluorescent protein (0.2 μg). One day after transfection, cells were washed with fresh medium and recordings were carried out within 24 h.

### Outside-out recordings and analysis
Recording currents are obtained with an RK-400 amplifier (BioLogic) using pClamp 10.5 software, digitized with a 1550 digidata (Axon instruments). Recording pipettes were obtained from thick-wall borosilicate glass (1.5 × 0.75 mm × 7.5 cm, Sutter Instrument) using a micropipettes puller (P-1000, Sutter Instrument) and fire-polished with a micro-forge (MF-830, Narishige) to be used at resistances

between 7 and 15 MΩ. Micropipettes were filled with internal solution (that contain in mM: 152 NaCl, 1 MgCl₂, 10 BAPTA, 10 HEPES; pH adjusted to 7.3 with NaOH solution, osmolarity measured at 335 mOsm). Extracellular solution (in mM: 152 NaCl, 1 MgCl₂, 10 HEPES; pH adjusted to 7.3 with NaOH solution and osmolarity was adjusted to 340 mOsm with glucose) was delivered by an automated perfusion system (RSC-200, BioLogic). Agonists' solutions are freshly made before sessions of recordings and are obtained with extracellular solution added with 1 to 100 μM of glycine (dissolved from a stock solution of 1 M in water). Acquisition of recordings was performed at the sampling of 20 kHz and low-pass filtered at 1 kHz (using the amplifier 5-pole Bessel filter). Openings are analyzed using Clampfit 10 software and currents were calculated by fitting the all-points histogram distributions of current amplitudes with the sum of two Gaussian curves. No further filtering is performed for the analysis.

### Molecular dynamics simulations
Molecular dynamics simulations of the zebrafish GlyR-α1 in four different conformational states were carried out: apo-closed (PDB:6PXD), Sy9-closed (PDB:6PXD with Sy9 docked), tau-closed (PDB:6PM3), and tau-open (PDB:6PM2). The zebrafish receptor shares more than 87% of sequence homology with the human receptor according to UNIPROT BLAST alignment. For all the studied systems, the protonation state at pH 7.0 was predicted using the Adaptive Poisson-Boltzmann Solver[53] (PropKa) via the web server ("https://server.poissonboltzmann.org/pdb2pqr"). Predicted protonation probabilities differed depending on the structure. To homogenize the systems, the most frequent predictions were selected and imposed on all systems: GLU316 was unprotonated, HIS125 protonated in Nδ, HIS217 protonated in Nδ, HIS327 protonated in Nδ, and HIS231 protonated in Nε. The structures were then embedded in a POPC bilayer and solvated with TIP3P water molecules using CHARMM-GUI Membrane Builder[54]. The CHARMM-36m forcefield[55] for the protein and CGenFF[56] for the taurine and strychnine were used for the all-atom molecular dynamics simulations. The simulations were performed using GROMACS 2021.4[57] with input files generated by CHARMM-GUI Membrane Builder for the minimization and equilibration. In short, 5000 steps of steepest descent were performed, then the system was equilibrated over 6 short simulations in the presence of atomic position restrains of decreasing strength, first in NVT, then in NPT ensemble, with the Berendsen thermostat and barostat. The temperature and the pressure were set to 300 K and 1 bar respectively. Finally, the equilibrated systems were carried on in production for 100 ns at a 2 fs timestep, in the NPT ensemble using V-rescale and Parrinello-Rahman as thermostat and barostat, respectively. In all cases, the LINCS algorithm was used to constrain the H bonds and the PME served to treat the long-range Coulomb interactions. For each system, we produced 5 replicas generated with different velocities from the equilibration.

MD simulations of the triple mutant used for the VCF experiments (K159W/Q235C/C57V, following the residue order in the PDB files) were also performed. Initial coordinates of the protein were extracted from the tau-closed (PDB: 6PM3) and tau-open (PDB: 6PM2) cryo-EM structures and the mutations were introduced using the CHARMM-GUI web server during preparation. The protocol followed to perform the simulations was the same as applied to the WT (see above).

### Trajectory analysis
The generated trajectories were analyzed using MDanalysis[58] through Python scripts. Notably, we computed the RMSD of the backbone atoms in the ECD, the TMD, and the structural core (i.e., the inner and outer beta-sheets in ECD, and the M1-M3 helices in TMD) relative to the initial coordinates from cryo-EM. Additionally, we computed the RMSF decomposed per residue and averaged over the 5 subunits of the

receptor. Analyses of the blooming and twisting angles[10] were carried out using WORDOM software (https://sourceforge.net/p/wordom/codehg/ci/default/tree/). For these calculations, only a subset of residues was considered, referred to as the core, which is more relevant to monitor the quaternary structure of the receptor. The selected core residues are 54–69, 73–86, 113–125, 142–154, 165–175, and 207–214 for the ECD and residues 225–234, 241–258, 265–288, and 297–322 for the TMD. Plots were produced using Matplotlib[59], and the visual representations of the protein with the open-source version of PyMOL[60] (Schrödinger).

### Docking
To explore strychnine binding to the various conformational states of the receptor, i.e., apo-closed, tau-closed, and tau-open, we extracted 1 snapshot per nanosecond from the trajectory generated by molecular dynamics. By doing so, we ended up with 5 (replica) × 100 (ns) = 500 (snapshots) for each system. The molecular docking of strychnine was then executed using QuickVina 2.1[61] with an exhaustiveness of 8 and a box of dimension 15 × 15 × 15 Å centered on the previously defined binding sites. For this purpose, we prepared the PDBQT inputs of the receptor using *prepare_receptort4.py* script from MGLTools[62]. Then, the positions of the 5 orthosteric binding sites of a given structure were computed on the fly as the center-of-mass of residues PHE223 of the (+)-subunit and PHE79 of the (−)-subunit using MDanalysis. Of note, for both tau-open and tau-closed, the receptor was simulated by Molecular Dynamics in the presence of taurine, but the docking of Strychnine was carried out without taurine.

### Reporting summary
Further information on research design is available in the Nature Portfolio Reporting Summary linked to this article.

## Data availability
Data supporting the findings of this manuscript are available from the corresponding author upon reasonable request. PDB structures referred to in the course of this work: 6PXD, 6PM2, 6PM3, 6PM6. Source data are provided with this paper.

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

## Acknowledgements

The work was supported by the ERC (Grant no. 788974, Dynacotine), by the "Agence Nationale de la Recherche" (Grant ANR-18-CE11-0015-01, Pentacontrol), Specific Grant Agreement No. 945539 (Human Brain Project SGA3), the doctoral school ED3C and the Foundation pour la Recherche Médicale (to Solène N. Lefebvre). The authors would like to thank Antoine Taly for their help in data interpretation, and Alexandre Mourot for critical reading of the manuscript.

## Author contributions

L.P. and A.C. contributed equally to the work. S.S. and S.N.L. set up and performed VCF experiments. L.P. performed single-channel experiments and some TEVC experiments and VCF experiments. M.G. designed and constructed the recording chamber. A.C. and P.M.R. performed MD and docking experiments. S.S., S.N.L., L.P., A.C., P.M.R., M.G., M.C., and P.-J.C. designed the study and analyzed the data. S.S., L.P., A.C., P.M.R., J.-P.C., M.C., and P.-J.C. wrote the manuscript. M.C., P.-J.C., and J.-P.C. acquired funding.

## Competing interests

The authors declare no competing interests.
