## [Peer Review File · Nature Communications]

Illumination of a progressive allosteric mechanism mediating the glycine receptor activationReviewers' Comments:

Reviewer #1:

Remarks to the Author:

Summary

This study set out to identify and characterize intermediate conformations during GlyR activation using a combination of VCF and MD. ECD conformational transitions that precede pore opening of GlyR and other (p)LGIC have been observed in functional and structural studies; this manuscript aims to connect structure and function and thus adds to understanding the progressive gating mechanism featuring conformational states that precede the primed and open states. The experiments are well executed, the figures are clear, and the manuscript reads nicely. Some of the conclusions are however not fully supported by data, the VCF and the MD could be better connected, and the insights gained are somewhat limited. The manuscript would highly benefit from further exploration of the intermediate state.

Main comments:

1) The difference in the concentration-response curve between the conformational- and the functional changes is striking and intriguing. If the conformational change observed with the fluorescence-quenching sensor here really relates to an ECD-wide 'early' intermediate conformation, it appears surprising that previous VCF recording could not pick up on it. What are the authors' thoughts on it? Data showing a similar phenotype when labelling in a different position would make this claim much stronger.

2) Further insights on how the described intermediate state can be (de-)stabilized by mutations would validate some of the assumptions made by the authors.

a. The N61D/N46D mutation shifts the glycine-elicited ΔI and ΔF curves to different degrees while activation with partial agonists approximately maintains the glycine-evoked shift between the curves. It would be interesting to hear the author's view on the mechanism behind the N61D/N46D mutation regarding its effect on glycine affinity and efficacy.

b. Related to the point above, it would be interesting to see if the introduction of known gain of function mutations (esp. outside of the orthosteric binding site or in the orthosteric binding site but explored with partial agonists) would bring the ΔI and ΔF curves closer together.

c. The manuscript currently bridges VCF data from a mutant with strongly shifted glycine sensitivity, with MD data of a WT-like channel. However, the relevance of the mutant to the WT and thus also the MD data remains questionable. MD simulations of the mutant could provide further insights.

3) Figure 5: Overlying the current and voltage traces gives the impression that the absolute size of the signals correlate. It would be better to separate the signals as done in the preceding figures to avoid confusion and better reflect the arbitrary choice in which the signal size is displayed.

4) While the manuscript leaves room for several intermediate states, the authors conclude that they all fall into the general category of pre-activated (ECD activated with TMD closed), solely based on the similarities observed in the quenching of the fluorescence signal. It is however possible that the glycine-strychnine combination and propofol only locally change the environment of the sensor in a similar way as (partial) agonists do, while the rest of the ECD could present itself vastly differently. This is especially true for propofol as it binds to the TMD of GlyRs and might thus present an alternative mechanism with which it differentially shifts the equilibrium to a more activated state (or direct activation). The strychnine data could also be better tied to previous insights from VCF experiments that include other domains of the ECD. In general, the possibility of alternative states that are not regarded as an 'early' pre-activated state should be addressed and the wording around it should be more carefully chosen to accommodate this possibility (e.g. short paragraph starting in line 202, Figure 7, but throughout).

5) Related to point 4, since propofol enhances glycine currents, an experiment showing a mixture of

propofol with increasing glycine concentration could be able to shed more light on the stabilization of the glycine-induced intermediate state.

6) Line 227 describes four unbinding events, while the caption in Figure S6 mentions three. In general, unbinding data should be shown in a plot to better illustrate its connection to blooming. For example, does unbinding in a specific subunit affect its blooming directly?

Minor comments:

- Line 123: The text describes the decrease in fluorescence as 2%, but in Fig. S3 the signal appears to be 0.2%.
- Figures 4A and B, change ΔI (black) and ΔF (cyan)
- For consistency, consider switching the left-right order of Figure 4, so the traces are on the left and the plots on the right.
- Add spaces between numbers and units: Fig 2c, Fig 5

Reviewer #2:

Remarks to the Author:

I believe this is a very well written manuscript that reports on a key advance of detecting a new intermediate state in the gating transition of an important class of eukaryotic ligand-gated ion channels. The study combines beautiful fluorescence probing to detect local conformational changes in the binding site with electrophysiology assays and molecular simulations to provide molecular-level hypotheses.

1. The systematic pattern detecting changes in fluorescence at glycine concentrations that are lower than those required for activation is interpreted as a new intermediate state. I don't have a problem with that nomenclature per se, but I wonder if it might be a bit too inspired by the tau-closed cryo-EM structures? Could it not alternatively be caused by glycine first only binding to a few subunits (lower concentration), which results in a fluorescence signal, but the actual activation and ion current would not happen until all five subunits have glycine bound? If anything, I find it more difficult to understand why lower glycine concentrations should elicit the same conformational change in all subunits, but that there should still be a second (full?) change when the glycine concentration increases?

2. For the plain MD simulations, I would not be quite as confident that four molecules unbinding in a (relatively) short 100ns simulation is evidence for weaker binding in the tau-closed state. The analysis of the C-loop is a good argument, but have the authors checked what the local resolution is in tau-open vs. tau-closed in this part of the structure, to ensure the difference is not simply caused by differences in the original structures?

3. Related to the point above, the unbinding events in the MD simulations are interpreted as tau-closed having lower affinity than tau-open, but in the subsequent docking studies there appear to be fewer frames from the tau-open simulations that are compatible with binding (compared to tau-closed), which seems to point in the opposite direction?

4. On line 253 (page 7), it says that the threshold -9.3 kcal/mol was used since it's within 10% of sty-closed). This seems to be a typo since stry-closed is reported as -12.1 kcal/mol; should it rather refer to apo-closed (with a value of -10.3)?

Ref1-1) The difference in the concentration-response curve between the conformational- and the functional changes is striking and intriguing. If the conformational change observed with the fluorescence-quenching sensor here really relates to an ECD-wide 'early' intermediate conformation, it appears surprising that previous VCF recording could not pick up on it. What are the authors' thoughts on it? Data showing a similar phenotype when labelling in a different position would make this claim much stronger.

Following the Reviewer's suggestion, to investigate other positions showing an "intermediate" phenotype, we introduced labeling at other positions nearby the K143W/Q219C pair, in the lower part of the subunit-subunit interface in the ECD. Among them, the mutant G181C produced a complex phenotype which was not easy to interpret. K143W/G181C and D141W/I188C (on both sides of the interface) gave current but no significant fluorescence changes. Therefore, we went back to the literature and reanalyzed the data collected by the Lynch lab. Since a major characteristic of the intermediate phenotype is a leftward shift of the ΔF curve as compared to the ΔI curve, we compared the various mutants of three key Pless et al papers (JBC 282, 36057–36067 (2007), JBC 284, 15847–15856 (2009), JBC 284, 27370–27376 (2009)). For each position of labeling, we plotted the $EC_{50}^{\text{fluorescence}}$ as a function of the EC_{50}^{current} .

In the plot above, the Pless mutants are colored black, and the one we present in the current version of our paper are colored cyan. The data points correspond to activation by glycine, unless when other ligands are specified. Mutants showing $EC_{50}^{\text{fluorescence}}/EC_{50}^{\text{current}}$ values lower than 1, which are indicative of an intermediate phenotype, are located below the diagonal. The plot shows that the vast majority of the Pless mutants do not display an intermediate phenotype. However, there is one exception, i.e., R271C activated by the partial agonist beta-alanine, which shows an intermediate phenotype. Using our VCF setup, we reproduced this observation and showed that R271C with taurine displays a $EC_{50}^{\text{fluorescence}}/EC_{50}^{\text{current}}$ ratio significantly lower to 1 (intermediate phenotype), while with glycine the ratio is close to 1. These data are now presented in a new figure 6 and subsection of the results (line 228) entitled “*The R271C mutant also shows a relative increase in fluorescence variation when activated by Taurine as compared to glycine.*” Our results show that an additional position on the vestibular side of the M2-M3 loop reports an intermediate-like phenotype, although the signal is weaker than that of K143W/Q219C.

In the Discussion (starting at line 376), we now compare the phenotypes of K143W/Q219C and R271C. Both mutants show a similar loss of function phenotype (10 to 20-fold rightward shift of the ΔI curve as compared to WT), with K143W/Q219C showing a stronger stabilization of the intermediate(s) conformations, as judged by the left shift of the ΔF curve compared to ΔI , as well as by the taurine and strychnine data (see discussion, lines 322-329). We speculate that the existence of an intermediate(s) state, which is very clear in K143W/Q219C and apparent in R271C, emerges as a result of allosteric mutations that were introduced to incorporate the quenching sensor (i.e. a loss-of-function phenotype due to a distinctive stabilization of the intermediate state relative to the resting state). We speculate that the absence of an intermediate phenotype signature when recording at other positions in previous analyses by us and the Lynch group might be due to the too low population of intermediate conformations in mutants displaying a WT-like EC_{50}^{current} . The plot above provides an illustration of this idea, since most mutants displaying an $EC_{50}^{\text{fluorescence}}/EC_{50}^{\text{current}}$ higher than 1 also show an EC_{50}^{current} in the WT range, while all mutants/conditions displaying an $EC_{50}^{\text{fluorescence}}/EC_{50}^{\text{current}}$ lower than 1 show an EC_{50}^{current} much higher than WT. The plot is now included as supplementary fig. 7.

This is now stated in the discussion line 386:

“We speculate that the K143W mutation, strongly stabilizing the intermediate(s) conformation, allows its robust monitoring by VCF. In retrospect, this might explain why the “intermediate” phenotype has rarely been observed in the previous VCF studies of the GlyR.”

Ref1-2) Further insights on how the described intermediate state can be (de-)stabilized by mutations would validate some of the assumptions made by the authors.

a. The N61D/N46D mutation shifts the glycine-elicited ΔI and ΔF curves to different degrees while activation with partial agonists approximately maintains the glycine-evoked shift between the curves. It would be interesting to hear the author's view on the mechanism behind the N61D/N46D mutation regarding its effect on glycine affinity and efficacy.

Mutations in the binding site are known to alter both the affinity and the efficacy of the agonist, and discriminating between these effects might require extensive investigations, possibly by kinetic analyses of single channel recording. The N61D/N46D mutant was just used as a control to show that the orthosteric site governs to a large extent the fluorescence variation. However, it produces a complex phenotype that may plausibly underly the shift of the ΔI and ΔF curves to different degrees. We think that investigating this point would require extensive kinetic studies that are beyond the VCF technique and the scope of this work.

b. Related to the point above, it would be interesting to see if the introduction of known gain of function mutations (esp. outside of the orthosteric binding site or in the orthosteric binding site but explored with partial agonists) would bring the ΔI and ΔF curves closer together.

To our knowledge, there is no described gain-of-function mutations at the level of the orthosteric binding site. To answer the referee, we recorded two new constructs where K143W/Q219C is combined with well characterized hyperekplexia mutations: the loss of function R271Q (top of M2, Langosch et al *FEBS Lett.* **336**, 540–544 (1993)) producing a rightward shift of the current dose-response curve, and the gain of function V280M (M2-M3 loop, Bode & Lynch *JBC* **288**, 33760–33771 (2013)) producing constitutive activation and a leftward shift of the current dose-response curve. K143W/Q219C/R271Q shows no current, but a ΔF curve shifted rightward (i.e. to higher glycine concentrations) more than 10-fold. K143W/Q219C/V280M also produces constitutive currents in the absence of glycine, a 100-fold increase in the sensitivity to glycine, but no variation of fluorescence. These mutants are presented on lines 170-179 and in supplementary figure 5.

While these new mutants do not show simultaneously ΔF and ΔI variations, the data are still compatible with the proposed intermediate(s) conformation. For R271Q, it is not surprising that the combination of two loss-of-function mutants (R271Q+K143W) yields a non-function channel. For V280M, a fraction of the population is constitutively in the active state, and we speculate that the rest would be in intermediate conformations rather than in the resting state. This would explain the lack of fluorescence variation observed by VCF (lines 332-334).

c. The manuscript currently bridges VCF data from a mutant with strongly shifted glycine sensitivity, with MD data of a WT-like channel. However, the relevance of the mutant to the WT and thus also the MD data remains questionable. MD simulations of the mutant could provide further insights.

Following the Reviewer's suggestion, we carried out new MD simulations of the GlyR triple mutant used in the VCF experiments (K143W/Q219C/C41V). Initial coordinates of the protein were extracted from the tau-closed (PDB: 6PM3) and tau-open (PDB: 6PM2) cryo-EM structures and the three mutations above were introduced using the CHARMM-GUI webserver. The simulation protocol was the exact same as for WT. For each conformational state (tau-open and tau-closed), two independent replicas of 100 ns were launched, and the simulation trajectories were compared with those collected for WT. The results are given in the Table below.

First, the analysis of the C α -RMSD shows that the mutant simulations are stable (C α -RMSD < 1.7 Å) and they deviate from the initial cryo-EM coordinated no more than WT. Second, analysis of the blooming and twisting angles, which are used to monitor the quaternary structure of the receptor, shows that the mutant simulations are WT-like, i.e., the closed state is more twisted and bloomed (i.e., expanded) than the open state. Remarkably, in the mutant simulations the twisting angle values perfectly match those in WT simulations and the blooming angle of the closed state is the same as in WT within errors. The only significant difference is in the blooming angle of the open state, which is slightly higher in the WT simulations. Based on this analysis, we conclude that the WT simulations presented in the first version of the manuscript are appropriate to explore the conformational dynamics of the triple mutant used in the VCF experiments.

System	RMSD (Å)	Blooming angle (°)	Twist angle (°)
Tau_open WT	1.62 +/- 0.02	10.61 +/- 0.30	16.51 +/- 0.34
Tau_closed WT	1.53 +/- 0.02	14.04 +/- 0.58	17.83 +/- 0.88
Tau_open Mutant	1.63 +/- 0.07	9.70 +/- 0.24	16.30 +/- 0.57
Tau_closed Mutant	1.64 +/- 0.14	15.20 +/- 0.47	17.54 +/- 0.31

Results on the C α -RMSD and the twisting and blooming angles for the WT and mutant simulations are now given in the supplementary Table S4. The time series of the twisting and

blooming angles for the mutant simulations are shown in the supplementary Fig. S9. The following sentence has been added to the Main Text (line 300):

“Although this analysis was carried out on the WT zebrafish receptor, we performed control MD simulations of the K143W/Q219C/C41V triple mutant in the tau-open and tau-closed states, giving similar results”

While doing the analysis of the mutant simulations, we found an error in the calculation of the blooming angle by the program Wordom (<http://wordom.sourceforge.net>). This error, which regards the sign of the blooming angle, affects the numerical results originally presented in Fig S6 but not the conclusion that the extracellular domain (ECD) of the apo or the strychnine-bound closed state is significantly more expanded (or bloomed) than that of the open- and closed-channel states bound to taurine. Correct results are now presented in Table S4 and the new version of Supplementary Fig. 8.

3) Figure 5: Overlying the current and voltage traces gives the impression that the absolute size of the signals correlate. It would be better to separate the signals as done in the preceding figures to avoid confusion and better reflect the arbitrary choice in which the signal size is displayed.

The referee is right, it has been done.

4) While the manuscript leaves room for several intermediate states, the authors conclude that they all fall into the general category of pre-activated (ECD activated with TMD closed), solely based on the similarities observed in the quenching of the fluorescence signal. It is however possible that the glycine-strychnine combination and propofol only locally change the environment of the sensor in a similar way as (partial) agonists do, while the rest of the ECD could present itself vastly differently. This is especially true for propofol as it binds to the TMD of GlyRs and might thus present an alternative mechanism with which it differentially shifts the equilibrium to a more activated state (or direct activation). The strychnine data could also be better tied to previous insights from VCF experiments that include other domains of the ECD. In general, the possibility of alternative states that are not regarded as an 'early' pre-activated state should be addressed and the wording around it should be more carefully chosen to accommodate this possibility (e.g. short paragraph starting in line 202, Figure 7, but throughout).

The referee is right to emphasize this point. The “intermediate” conformations may be different in the different pharmacological conditions, and the present VCF data cannot distinguish between them. Still, the MD simulation of the cryo-EM Tau-closed state provides a reasonable

and unifying explanation of the intermediate states observed with (partial) agonists and the glycine-strychnine mixture. We agree that data at high propofol concentration may underly a different mechanism. To be more explicit about this we added in the result section line 222:

“However, it is possible that the glycine-strychnine combination and propofol alone change the environment of the sensor in a similar way as partial agonists and agonists do, giving similar ΔF signature, while the rest of the ECD could present different conformations. This is especially possible for propofol alone as it binds to the TMD of GlyRs and might thus present an alternative mechanism.”

From MD simulation, we conclude in line 302:

“The simulation results support the conclusion that the intermediate conformation(s) revealed by VCF, concerning both (partial) agonists and glycine-strychnine combinations, is consistent with a highly dynamic cluster of states as described by MD simulation started from the cryo-EM tau-closed structure.”

In the discussion, we state in line 370

“Although we cannot exclude the existence of alternative intermediate states in different pharmacological conditions e.g. involving changes in the local environment of the fluorescence sensors, the MD simulations of the cryo-EM Tau-closed state provides a reasonable and unifying explanation of the intermediate states recorded with (partial) agonists and the glycine-strychnine mixture.”

5) Related to point 4, since propofol enhances glycine currents, an experiment showing a mixture of propofol with increasing glycine concentration could be able to shed more light on the stabilization of the glycine-induced intermediate state.

Following the referee suggestion, we performed a full dose-response curve of glycine in the presence of a 100 μM concentration of propofol (eliciting alone no ΔI or ΔF). We found that propofol potentiates by $21 \pm 7\%$ the maximal currents and shifts by 8-fold the ΔI curve to lower concentrations. Interestingly, propofol does not affect significantly the ΔF curve. Propofol thus does not alter the transition mediating the fluorescence change, but specifically facilitates the “downward” process of activation. It is noteworthy that the glycine+propofol condition is unique, since the intermediate phenotype is observed while the $\text{EC}_{50}^{\text{current}}$ is in the WT-range (see plot above). Data are presented in Fig. 4C and in the results line 210-216.

6) Line 227 describes four unbinding events, while the caption in Figure S6 mentions three. In general, unbinding data should be shown in a plot to better illustrate its connection to blooming. For example, does unbinding in a specific subunit affect its blooming directly?

As stated in line 227, we observed four taurine unbinding events from tau-closed and none from tau-open in the WT simulations. To better illustrate this phenomenon, we monitored the all-atom RMSD from cryo-EM for each taurine molecule over time and without optimal superimposition of the atomic coordinates of the ligand to account for both translational and rotational motions. The results clearly show two unbinding events from replica 1, one in replica 2, and one in replica 4, while no taurine unbinding is observed in replica 3 and replica 5. In addition, this analysis shows that two taurine unbinding have been captured in one replica of the mutant simulations. These results are now included as supplementary Fig. S10.

In addition and following the Reviewer's suggestion, the correlation between taurine unbinding and the quaternary structure of the receptor was investigated by comparing the time series of the blooming angle in simulations with taurine unbinding (e.g., replica 1) with simulations with no taurine unbinding (e.g., replica 5). The results show that with taurine unbinding (replica 1) the blooming angle is unstable and increases to a value of 18° at the end of the simulation. In sharp contrast, w/o unbinding (replica 5) the blooming angle stabilizes about a value of 13° (see Figure below). More generally, we find that the blooming angle averaged over the last 20 ns in both WT and mutant simulations is >14.8° when one or more taurine ligands unbind, whereas it is <13.3° when all taurine ligands remain bound (see Table below). We conclude that receptor's blooming is enhanced when less than five taurine ligands are bound, which indicates that agonist or partial agonist binding have an influence on the quaternary structure of GlyR as discussed previously (<https://doi.org/10.1016/j.neuropharm.2014.12.006>).

System	Replica	Unbinding events	Blooming angle (°)
Tau-Close Wild Type	1	2	17.44 +/- 0.64
	2	1	15.92 +/- 0.52
	3	0	13.27 +/- 0.42
	4	1	14.87 +/- 0.42
	5	0	12.77 +/- 0.42
Tau-close Triple Mutant	1	0	12.86 +/- 0.43
	2	2	18.49 +/- 0.87

Minor comments:

- Line 123: The text describes the decrease in fluorescence as 2%, but in Fig. S3 the signal appears to be 0.2%.
- Figures 4A and B, change ΔI (black) and ΔF (cyan)
- For consistency, consider switching the left-right order of Figure 4, so the traces are on the left and the plots on the right.
- Add spaces between numbers and units: Fig 2c, Fig 5

These four suggestions have been taken into account and corrected in the revised manuscript.

Ref2-1. The systematic pattern detecting changes in fluorescence at glycine concentrations that are lower than those required for activation is interpreted as a new intermediate state. I don't have a problem with that nomenclature per se, but I wonder if it might be a bit too inspired by the tau-closed cryo-EM structures? Could it not alternatively be caused by glycine first only binding to a few subunits (lower concentration), which results in a fluorescence signal, but the actual activation and ion current would not happen until all five subunits have glycine bound? If anything, I find it more difficult to understand why lower glycine concentrations should elicit the same conformational change in all subunits, but that there should still be a second (full?) change when the glycine concentration increases?

The referee is right. The change in fluorescence indeed is complete at subsaturating glycine concentrations, underlying partial occupation of the orthosteric site but a quasi-symmetric reorganization at the ECD-TMD interface (nearby the fluorescent sensor). We suggest that occupation of few orthosteric sites by glycine would reorganize not only the concerned interface, but also the other subunit-subunit interface of the pentamer possibly via a cooperative mechanism. This would lead to a symmetrical reorganization of the five quenching pairs at the ECD-TMD interface without full orthosteric site(s) occupation. Upon binding of more glycine molecules, a second cooperative step might occur, eventually leading to a concerted motion of the M2 helices to open the ion pore.

The Tau-closed structure being fully symmetric, it cannot account per-se for the VCF data, especially concerning glycine-strychnine mixture. However, the simulation data show that the Tau-closed structure visits asymmetric conformations with a major reorganization of only a

fraction of the orthosteric sites, a feature that can plausibly explain not only the glycine strychnine data, but also the glycine data a low concentration. Our view is that, while the middle and upper ECD might be strongly asymmetric particularly at the level of the orthosteric site, the lower ECD and possibly the TMD would keep a more symmetric organization. It is noteworthy that asymmetric reorganizations of the ECD coupled to symmetric reorganization at the TMD is common in heteromeric pLGICs, well documented on the muscle nAChR from the latest Cryo-EM work by Ryan Hibbs and Hugues Nury, for instance.

2. For the plain MD simulations, I would not be quite as confident that four molecules unbinding in a (relatively) short 100ns simulation is evidence for weaker binding in the tau-closed state. The analysis of the C-loop is a good argument, but have the authors checked what the local resolution is in tau-open vs. tau-closed in this part of the structure, to ensure the difference is not simply caused by differences in the original structures?

Following the Reviewer's suggestion, we analyzed the local resolution of residues lining the orthosteric site by comparing the β -factors of the C_{α} carbons of the C-loop in the cryo-EM structures used for MD. We found that in tau-open (6PM2) the β -factors in this region of the protein range from 74.29 to 84.85 \AA^2 , while in tau-closed (6PM3) they vary from 35.78 and 41.78 \AA^2 . Therefore, residues in the orthosteric site appear as better resolved in the tau-closed state, which shows multiple taurine unbinding during MD. We conclude that taurine unbinding is a genuine simulation result that is not related to the resolution of the original structures.

3. Related to the point above, the unbinding events in the MD simulations are interpreted as tau-closed having lower affinity than tau-open, but in the subsequent docking studies there appear to be fewer frames from the tau-open simulations that are compatible with binding (compared to tau-closed), which seems to point in the opposite direction?

The results in Fig. 7B show that that only a minor fraction of GlyR conformations sampled in the tau-open state are compatible with strychnine (and not taurine) binding. This observation is unrelated to the affinity for taurine, which is a much smaller ligand that fits the orthosteric pocket both in the open and closed states as shown by cryo-EM.

Reviewers' Comments:

Reviewer #1:

Remarks to the Author:

The revised version of this manuscript sufficiently addresses my comments. ¹¹_{SEP}The additional experiments (particularly the R271C data, effects of taurine unbinding on blooming, and the additional propofol data) have made the manuscript stronger. It is unfortunate that the other positions didn't lead to any current (+R271Q) or fluorescence (+V280M).

Just a few small comments:

- Given that the the authors don't see any change in fluorescence for +V280M and speculate that it is because the mutant already populates intermediate state(s), wouldn't you expect to see a signal when applying strychnine?

-Line 172: R219Q should be R271Q

-Wording in Line367/387: ...intermediate conformation(s), allowing ...

Reviewer #2:

Remarks to the Author:

I appreciate all the extra work carried out by the authors, and definitely believe the current version of the manuscripts should be published. The authors' suggestion that the fluorescence pattern observed is because occupation of only a few orthosteric sites would still lead to transitions at other subunit interfaces is a clear hypothesis that should be eminently testable in the future, even if it is not conclusively proven here.